# Immobilization of Lipases Using Poly(vinyl) Alcohol

**DOI:** 10.3390/polym15092021

**Published:** 2023-04-24

**Authors:** Nadia Guajardo

**Affiliations:** Programa Institucional de Fomento a la Investigación, Desarrollo e Innovación, Universidad Tecnológica Metropolitana, Santiago 8940000, Chile; nguajardo@utem.cl

**Keywords:** lipases, polyvinyl alcohol, immobilization of enzymes, supports, alginate, chitosan, hydroxypropylmethylcellulose

## Abstract

Lipases are very versatile enzymes because they catalyze various hydrolysis and synthesis reactions in a chemo-, regio-, and stereoselective manner. From a practical point of view, immobilization allows the recovery and stabilization of the biocatalyst for its application in different types of bioreactors. Among the various support options for immobilizing lipases is polyvinyl alcohol (PVA), which, when functionalized or combined with other materials, provides different characteristics and properties to the biocatalyst. This review analyzes the multiple possibilities that PVA offers as a material to immobilize lipases when combined with alginate, chitosan, and hydroxypropylmethylcellulose (HPMC), incorporating magnetic properties together with the formation of fibers and microspheres. The articles analyzed in this review were selected using the Scopus database in a range of years from 1999 to 2023, finding a total of 42 articles. The need to expand knowledge in this area is due to the great versatility and scaling possibilities that PVA has as a support for lipase immobilization and its application in different bioreactor configurations.

## 1. Introduction

Enzymes are biological biocatalysts responsible for catalyzing biochemical reactions in living beings. There are different classifications depending on the reaction they catalyze [1]. Within the group of hydrolases are lipases, whose natural function is the hydrolysis of triglycerides [2,3]. From a practical point of view, lipases (triacylglycerol hydrolases, EC 3.1.1.3) are important biocatalysts in several applications, such as the synthesis of chiral drug intermediates [4] and nutraceutical lipids [5], bioconversion of oils and fats [6], and production of biodegradable polymers based on the ability of lipases for esterification, transesterification [7,8], aminolysis [9], and hydrolysis reactions [10]. The sources of lipases are found mainly in animals, plants, and microorganisms, where they play an essential role in their metabolism [11,12,13].

Some enzymes are expensive and, due to their biological nature, are susceptible to being inactivated under aggressive process conditions. This represents a challenge for its industrial application that can be overcome using immobilization techniques [3,14,15]. Different immobilization methodologies can be classified into physical, physical/chemical, and chemical methods [16,17,18], as shown in Figure 1. Among the physical techniques is adsorption, which consists of binding the enzyme to the support utilizing weak bonds such as those of van der Waals and hydrophobic support/enzyme interactions, such as the binding of lipases to hydrophobic supports [16,17,19]. Most lipases have a lid covering the enzyme’s active site, which opens in the presence of hydrophobic surfaces for catalysis. This mechanism is called the “interfacial activation mechanism,” which is applied in the immobilization of lipases by adsorption on hydrophobic surfaces [19]. A disadvantage of this methodology is the possibility of leaching of the enzyme from the support due to the weakness of their interaction.

Chemical methods, unlike adsorption methods, prevent the leaching of the enzyme from the support, increasing the stability of the biocatalyst. Among the chemical methods are the union by covalent bonds of the enzyme to the support [16,20] and the crosslinking enzyme aggregates (CLEA) that consist of the covalent union of the enzyme molecules between them, using a chemical crosslinking agent [15,21]. This methodology does not require support [15,21]. However, to improve its mechanical properties and apply them in enzymatic bioreactors, the CLEAS must be immobilized on supports [22,23].

The entrapment consists of the confinement of the enzyme within a polymeric network of gels and fibers, or the encapsulation consists of the confinement of the enzymes within a matrix that can be made of chitosan [24,25], alginate [24,26], or polyvinyl alcohol [27,28].

The physical/chemical methodology combines physical and chemical methods. An example is the entrapment of CLEA in polymers [22,23,29].

The methodologies of immobilization by entrapment and the combined procedure of physical and chemical methods can use polyvinyl alcohol to form fibers or gels and give the biocatalyst operational stability for its application in bioprocesses. Table 1 indicates the advantages and disadvantages of the immobilization methodologies and supports most used for the immobilization of lipases.

Polyvinyl alcohol (PVA) is a polymer frequently used as a matrix for immobilizing various enzymes and cells because of its easy availability, low price, and hydrophilic character and hydroxyl groups on the surface capable of chemical reaction [30,31]. However, its functionalization and combination with other materials can improve the biocatalyst’s catalytic properties, such as operational stability and scalabilities, like the commercial product made of PVA called LentiKats^®^ with good mechanical properties for its application in the immobilization of enzymes and cells [23,29,32].

Considering its versatility and the different options of PVA as a support in immobilization, this review reports the analysis of its main applications in lipase immobilization. The articles analyzed in this review were selected using the Scopus database in a range of years from 1999 to 2023, finding a total of 42 articles distributed per year, as shown in Figure 2.

## 2. Immobilization of Lipases on Crosslinked PVA Matrices

Due to its physical/chemical properties, PVA has been used in various applications, including the immobilization of lipases by physical methods, such as adsorption, and chemical methods, such as covalent binding. For example, the covalent immobilization of pancreatic lipase on crosslinked PVA was investigated by Guliz Ak et al. 2014 [33] and Taylan et al. 2010 [34]. The PVA matrix was crosslinked using the adipoyldichloride reagent [34,35] to subsequently immobilize the enzyme, which was subjected to pretreatment with different oils (olive oil, soybean oil, hazelnut oil, and corn oil), to favor the open conformation of the lipase active site and thus increase its activity during catalysis (Figure 3). This pretreatment consisted of incubating the enzyme with the oils for 5 min under constant agitation before enzyme-support covalent immobilization. The best result was obtained with the pretreatment of 20% olive oil with an immobilization yield per activity (IY) of 24% and an immobilization yield per protein (BY) of 44%. It is important to note that without this pretreatment, the IY was 3% and the BY 37%. This indicates that this immobilization negatively modulates the enzyme’s active site, and the pretreatment allows the active site to be less affected [33,34].

Another example is the immobilization by adsorption of *Candida rugosa* (CR) lipase on a PVA support crosslinked with epichlorohydrin and esterified with linear fatty acids of different chain lengths (*Cn*-CL-PVA), as detailed in Figure 4. The immobilization of CR lipase on different supports, such as Celite and C*n*-CL-PVA, was compared. The best results in terms of specific activity (U/g support) were those matrices esterified with fatty acids with a chain size between 8 and 12, reaching specific activities over 180 and this activity being 140 times greater than that of CR lipase immobilized on Celite [36]. This behavior indicates that the hydrophobicity of the support stabilizes the lipase’s open conformation, increasing the biocatalyst’s activity [36].

## 3. Immobilization of Lipases on PVA Hybrid Matrices

### 3.1. Immobilization on PVA/Alginate

Alginates are among the most used polymers due to their mild gelling properties and nontoxicity, improving enzyme stability and functional properties. For example, PVA hydrogels with calcium alginate are widely used as matrices to immobilize enzymes and cells [37]. This system combines the advantages of the two supports by itself. PVA has the advantage of having a lower resistance to mass transfer, while alginate avoids agglomeration between the beads and provides more excellent mechanical resistance to the matrix. An example is the immobilization of the lipase of Pachira aquatica using beads of calcium alginate (Alg) and polyvinyl alcohol) (PVA). The work aimed to evaluate the reuse and stability of the immobilized enzyme compared to the free enzyme. The immobilized lipases were more stable than the free enzyme, retaining 60% of their activity at 50 °C for 4 h. The immobilized enzyme was reused many times, with a 50% decrease in activity after five cycles [38].

*Thermomyces lanuginosus* (TL) lipase for biodiesel production was immobilized on PVA-alginate beads. The protein immobilization yield was approximately 94%. However, from the point of view of activity, the immobilization produced damage to the conformation of the enzyme, causing a reduction in activity concerning the free enzyme. The yield of biodiesel produced was 92% [39]. A variant of the previous methodologies is the immobilization of *Candida rugosa* in a PVA-alginate matrix crosslinked with boric acid [40]. This immobilization methodology supports an amphiphilic nature to be used in aqueous or hydrophobic systems. The performance of the biocatalyst was evaluated in the synthesis of ethyl hexanoate in isooctane. The thermal stability of the enzyme increased 10-fold with immobilization. It also retained almost 100% activity when reused in 10 reaction cycles and performed better than the free enzyme in the synthesis of ethyl hexanoate [40].

Another variation to this methodology is the immobilization of lipase (from the porcine pancreas) on PVA-alginate nanofibers [41]. The nanofibers were prepared by electrospinning. The results showed that the immobilization of the enzyme on PVA-alginate nanofibers at 60 °C for 40 min was 80% higher than the free enzyme and 70% higher than the free enzyme at 70 °C for 40 min. In the reuse of the biocatalyst, the enzyme immobilized in PVA-Alginate nanofibers retained 60% of its activity after 14 reaction cycles [41].

The need to provide the enzyme with a microenvironment that increases its operational stability and storage stability has motivated the development of new immobilization methodologies that combine halloysite nanotubes (HNTs)/lipase/PVA/alginate/ [27]. The immobilization methodology consists of the immobilization of *Candida rugosa* (CR) lipase on HNTs and then an entrapment of the CR/HNTs biocatalyst in a PVA/alginate hydrogel matrix [27]. The biocatalyst was 80% more stable than the free enzyme at 60 °C. Concerning its storage stability, the biocatalyst retained 68% of the activity concerning its initial activity for 30 days at four °C [27]. A variation of this methodology is the immobilization by adsorption of the CR lipase on SiO_2_ nanoparticles and then the entrapment of the CR/SiO_2_ biocatalyst in PVA/alginate hydrogels (Figure 5) [42]. The thermal stability of the biocatalyst showed an increase in its relative activity of 75% concerning the free enzyme. The stability in storage at 4 °C for 30 days, the biocatalyst showed an increase in its relative activity of 70% concerning the free enzyme and 30% concerning the lipase immobilized on SiO_2_ [42].

### 3.2. Immobilization on PVA/Chitosan

Applying PVA hybrid materials improves its properties, and just like alginate, chitosan (CH) also fulfills that purpose. CH is a biopolymer linked by beta 1,4 bonds to the polysaccharide matrix, providing PVA mechanical resistance. Due to the presence of free amino groups in each chitosan unit, it offers various possibilities of immobilization for lipases. An example is the immobilization of *Burkholderia cepacia* (BCL) lipase in a PVA/CH matrix, evaluated as a biocatalyst in optimizing esterification to obtain citronellyl laurate ester. The independent variables considered were the temperature, the concentration of the substrates, the amount of biocatalyst, and the pressure. At the same time, the response variable was yield. The highest yield was 94% using the operational conditions of 8 mmol Citronellol, 16 mmol laureate, 200 mg (CH/PVA/CH), 9.2 MPa, and 48 °C [43].

Another variant of applying these matrices in the immobilization of lipases is the covalent Immobilization of CR lipase on a PVA/CH matrix [44]. The immobilization methodology consisted of the preparation of the PVA/CH matrix to subsequently activate the surface of this matrix with glutaraldehyde, followed by the addition of the enzymatic solution. From the point of view of operational stability in sequential batches, using as a reaction model for phenethyl ester synthesis, the reaction conversion decreased by only 10% after the sixth reaction cycle. When comparing the performance of the PVA/CH/CR: GLU biocatalyst with the free enzyme and immobilized lipase alone on the PVA/CH/CR matrix, it was obtained that the PVA/CH/CR biocatalyst: GLU was 5% more stable than the PVA/CH/CR biocatalyst and 100% more stable than the free enzyme [44].

PVA/CH-based supports have also been applied to the immobilization of *Rhizopus oryzae* lipase (ROL) to catalyze the hydrolysis of triglycerides to produce monoglycerides [45]. The immobilization methodology consisted of preparing the PVA/CH matrix, which was later functionalized using glutaraldehyde (GLU) to finally covalently immobilize the ROL lipase to the support (PVA/CH/ROL: GLU). The biocatalytic membrane obtained was used in a membrane bioreactor to carry out the hydrolysis reaction, as detailed in Figure 6. The membrane bioreactor was used in a two-phase system, and the membrane was reused in 9 reaction cycles, reaching a yield of 32 to 50% [45].

Another example of lipase immobilization on PVA/CH matrices for application in membrane bioreactors is the immobilization of *Thermomyces lanuginosus* (TL) lipase (PVA/CH/TL). The membrane was characterized by SEM, finding that it was a continuous membrane without holes or breaks. The biocatalyst reuse was tested in 31 reaction cycles, using the hydrolysis of para-nitrophenyl palmitate (PNP) as a reaction model. The results showed a loss of enzyme activity of only 44% [46].

### 3.3. Immobilization on PVA/Hydroxypropyl Methylcellulose (HPMC)

Hydroxypropyl methylcellulose (HPMC) is a non-ionic cellulose ether obtained by cellulose alkalization, etherification, neutralization, and washing. HPMC has good thickening, dispersing, emulsifying, film-forming properties, etc. At the same time, the PVA matrices give the materials high resistance, flexibility, excellent adhesive properties, and resistance to solvents. The interesting physicochemical properties of both materials allow for obtaining good support for forming a film to immobilize enzymes.

An example is the immobilization of *R. oryzae* lipase on PVA/HPMC matrices. The biocatalyst preparation consisted of synthesizing the PVA/HPMC membrane and adding the lipase solution, which adhered to the matrix by adsorption [30]. The biocatalyst was evaluated in the esterification reaction of benzyl alcohol with vinyl acetate. The experimental results showed that the synthesis catalytic activity of the immobilized lipase was four times that of the free enzyme. The immobilized lipase was recycled in four consecutive cycles with a 32% activity loss and exhibited good storage stability for 90 days [30].

Another investigation reports the immobilization of *Candida cylindracea* (CCL) on a PVA/HPMC matrix to catalyze the synthesis of propyl benzoate. The biocatalyst reuse was evaluated in a sequential batch system, decreasing its initial activity by 40% at the fourth reaction cycle [28].

PVA/HPMC binary supports have also been used for the immobilization of *Burkholderia cepacia* lipase (BCL) to catalyze the kinetic resolution of 3- phenoxy-propane-1,2-diols (Figure 7) [47]. This immobilization technique improved the enzyme activity and its operational stability. The reaction conversion was 45 to 50% with an enantiomeric excess of 99%. The operational stability from the point of view of the enantiomeric excess was evaluated in a bioreactor by repeated batches, in which almost no decrease in chirality was evidenced at the end of the tenth cycle [47].

Another investigation by the same research group applies the PVA/HPMC biocatalyst to catalyze the synthesis of n-amyl acetate from n-amyl alcohol, a by-product of the sugar industry [48]. The conversion of the reaction was optimized, reaching a maximum value of 99.5% at a temperature of 50 °C, a biocatalyst load (PVA/HPMC/BCL) of 40 mg, and a molar ratio of acyl donor to alcohol of 2:1. The synthesis reaction was also carried out in a continuous membrane bioreactor with a residence time of between 12 and 60 min, reaching conversions of 5.6 to 22%. The operational stability was evaluated in five reaction cycles, where the biocatalyst showed no activity loss [48].

### 3.4. Immobilization on Different PVA Hybrid Supports

*Hypromellose* (HY) is a cellulosic water-soluble polysaccharide composed of hundreds of β-1-4 linked glucose units with excellent film-forming properties, flexibility, texture, biodegradability, and stability. Furthermore, the presence of free OH groups in HY provided an enhanced immobilization effect. The combination of HY and PVA generates a matrix of simple preparation for the immobilization of enzymes. The immobilization methodology consisted of a one-pot process in which HY/PVA and the enzyme are mixed to produce a film. The benefits of this immobilization methodology are that it is simple and does not require the application of crosslinking agents and that the entrapment protects the enzyme from the reaction medium, increasing its stability [49].

An example is the immobilization of BCL lipase in HY/PVA, which was incubated in different organic solvents, and its half-life (*t*) and inactivation constant (K_D_) were determined. The biocatalyst showed a high t (822 h) in the presence of organic solvents compared to polar solvents such as acetone (347 h). Furthermore, the K_D_-value was lower in cyclohexane (0.843 × 10^−^^3^) than in acetone (1.997 × 10^−^^3^), indicating better stability in the non-polar solvents. The biocatalyst HY/PVA/BCL was evaluated in the catalysis of synthesizing phenylbutyrate, a natural constituent of essential oils, and has wide industrial applications. After three h at 44 °C, a conversion of 99% was reached. The enzymatic activity of the HY/PVA/BCL biocatalyst was six times higher than that of the soluble enzyme, while the operational stability (reuse of the biocatalyst) was eight times higher than that of the free enzyme [49].

Another material used in combination with PVA is polysiloxane (SiO_2_). In preparing these matrices, the sol-gel methodology is used, which consists of three stages: Hydrolysis, subsequent polycondensation of tetraethoxysilane, and the formation of a hybrid SiO_2_/PVA matrix [50]. An example of the application of these supports in enzymatic bioreactors is in the glycerolysis of babassu oil by *Burkholderia cepacia* lipase immobilized on SiO_2_/PVA particles in a continuously packed bed reactor. The biocatalyst preparation consisted of synthesizing the SiO_2_/PVA support and then immobilizing the enzymatic solution. The reaction operation in a solvent-free system was carried out at 25 °C at a fixed space-time of 9.8 h using different molar ratios of babassu oil to glycerol to evaluate productivity and selectivity. The highest productivity and selectivity were 52.3 ± 2.9 mg g^−1^ h^−1^ and 31.5 ± 1.8%, respectively [51]. Paula et al., 2008, also studied the immobilization of porcine pancreatic lipase on SiO_2_/PVA matrices. The biocatalyst was evaluated in producing biodiesel from babassu oil and ethanol, efficiently converting triglycerides into alkyl ester fatty acids, reaching yields of 75–95% [52].

Sericin (SS) is a silkworm Bombyx mori protein that acts as a natural glue that attaches silk fibers (fibroin). During silk production, SS is removed and discharged. The molecular weight of SS ranges from 24 to 400 kDa,12 and serine (32.3%),13 aspartic acid (16.71%), and glycine (13.49%) are the primary amino acids. This biomaterial enriches polymer mixtures reducing environmental damage due to its degradability [53]. This material, in combination with cassava starch and PVA, has been used to immobilize *Botryosphaeria ribis* (Br) lipase using a sandwich adsorption methodology. The support preparation consisted of synthesizing the SS/PVA/CS film and then the adsorption of the enzymatic solution on the SS/PVA/CS material to form the SS/PVA/CS/Br biocatalyst. For optimal conditions, the yield in ethyl oleate was 95% for the immobilized enzyme. The maximum yield was obtained at 49 °C, molar ratio oleic acid: Ethanol of 1:3, 1.25 g lipase film or 50 U (1.03 ± 0.03 mg protein), and 30 h of reaction. Even after seven use cycles, immobilized lipase showed a 52% reduction in ester yield [53].

Another material that is used in combination with PVA for the immobilization of lipases is Polysulfone. Polysulfone (PS) are high-performance polymers whose characteristic is stability at high temperatures. Gupta et al., 2009 [54] studied the immobilization of *Candida rugosa* lipase (CRL) on a PS membrane modified with PVA (PS/PVA). The lipase was immobilized using covalent immobilization by the crosslinking agent glutaraldehyde (Glu) and adsorption. The support and immobilized lipase were characterized using X-ray diffraction (XRD) and Scanning Electron Microscope (SEM). Through XRD, it was possible to distinguish a peak belonging to the presence of lipase, and through SEM, it was possible to observe the presence of the enzyme in the matrix. The reuse of the biocatalyst PS/PVA/CRL:glu showed better operational stability with a decrease of 10.7% compared to the biocatalyst PS/PVA/CRL (with a reduction of 33.3%) after five reaction cycles [54].

Polymer modification has received much attention due to the properties and characteristics it can contribute to materials. Within the chemical methods for modifying polymers, grafting is one of the most interesting methods, imparting a variety of new functional groups to the polymer [55]. One of the methodologies for polymer grafting is so-called photografting is performed by irradiating it in a solvent containing selected monomers with appropriate radiation sources. Ultraviolet energy (UV) has been extensively applied for surface graft polymerization of polymers with a photoinitiator or photosensitizer, such as benzophenone (BP). This methodology was used for graft copolymerization of glycidyl methacrylate (GMA) onto polyvinyl alcohol (PVA) using benzophenone (BP) as initiator and form support (GMA/PVA) for the immobilization of porcine pancreatic lipase (PL). The thermal stability of the GMAIL/PVA/PL catalyst was 30% higher than the stability of the free enzyme and 40% more stable than the free enzyme when stored for 25 days [55].

Hal is a type of mineral clay and aluminosilicate similar to kaolin with the same chemical formula Al_2_Si_2_O_5_(OH)_4_. Hal nanotubes are ultra-small tubes whose diameter is generally less than 100 nm. The lengths of these nanotubes range from 500 nm to 1.2 μm. Aluminum, silicon, and hydrogen are the critical components of Hal nanotubes (HNTs) [56,57]. These nanotubes are ideal for enzyme immobilization because, being hollow and small in size, they have a large specific surface area and are highly biocompatible [56,57]. Combining HNTs and PVA gives special enzyme support because nanotubes, despite their enormous specific surface area, are microscopic and difficult to recover from the reaction medium. This problem is solved by entrapping them in matrixes of PVA. This is the case of the immobilization of *Candida rugosa* lipase (CRL) on a HNTs/CRL/PVA/Alginate matrix. The lipase adsorption on the HNTs nanotubes was optimized, and the nanotubes with the immobilized enzyme were subsequently confined in the PVA/Alginate matrices. The thermal stability of the HNTs/CRL/PVA/Alginate biocatalyst at 70 °C was 40% higher than the free enzyme and 35% higher than the enzyme immobilized in HNTs, showing that the entrapment in PVA/Alginate provides more stability to the biocatalyst [27].

## 4. Immobilization of Lipases in PVA Microspheres and LentiKats^®^

PVA microspheres have been used to entrap pharmaceutical and food bioactive compounds. However, due to their attractive properties, PVA microspheres have also been used for enzyme immobilization. The characteristics that make microspheres interesting for their application as supports are: Their mechanical stability and their spherical or oval shape, which allow their application in different configurations of bioreactors, and their crosslinked three-dimensional matrix prevents enzyme leaching. An example is the immobilization of *Rhizomucor miehei* (RM)lipase in crystalline PVA microspheres to catalyze the transesterification of soybean oil to obtain fatty acid ethyl esters (FAEE) [58]. The enzymatic activity of biocatalysts depends on the degree of crosslinking of the PVA matrix. The authors note that this activity loss may be due to the enzyme not being immobilized in its active form. However, due to the degree of crosslinking of the material, mass transfer problems could have affected the activity of the biocatalyst. The percentage of ethyl esters obtained with the enzyme immobilized on PVA microspheres was 45% higher than that obtained with the free enzyme. Based on the material’s spectrographic data, the biocatalyst’s good performance is explained by the interaction of the intermolecular network of hydrogen bonds of the PVA matrix with the residual amino acids on the enzyme surface [58].

An interesting methodology developed by Piacentini et al. (2017) is the immobilization of *Candida rugosa lipase* (CRL) in PVA microspheres through a membrane emulsification process followed by crosslinking with glutaraldehyde (GA) [59]. The microsphere preparation methodology consisted of a two-stage process. The first stage consisted of preparing the microspheres, where the dispersed phase formed by 15% PVA passes through the membrane’s pores, whose exterior is formed by a continuous step of isooctane, as shown in Figure 6. After creating the PVA microspheres, the lipase is physically immobilized by adsorption and then covalently by crosslinking using glutaraldehyde (Figure 8).

Several techniques have been developed for bead production from viscous fluids [32]. However, most of these techniques have a limitation from the practical point of view for their industrial scaling. Compared to these methodologies, LentiKats^®^ (LK) technology uses the gentle technique of gelation and a different method of preparation already scaled to industrial use. These particles combine the benefits of both large and small beads. The reason for this is the lens-shaped structure of the carrier, which has a diameter of 3 to 4 mm and a thickness of only 200 to 400 μm. An example of the application of PVA LentiKats^®^ (LK) is in the entrapment of immobilized *Candida antarctica* lipase B in the form of crosslinked aggregates (CLEA-CALB-LK) [23]. In this work, the operational stability of biocatalysts was evaluated when reusing them in many reaction cycles. It should be noted that at the end of the sixth cycle, the CLEA-CALB-LK biocatalyst reached a 25% higher conversion than the CLEA-CALB biocatalyst, indicating that the entrapment in PVA of the CLEA-CALB biocatalyst provides stability and mechanical resistance to the biocatalyst for its operation of different types of bioreactors [23].

## 5. Immobilization of Lipases on Magnetic PVA Supports

The recovery of the biocatalyst from the reaction medium in a simple and fast way has been the subject of research within bioprocessing engineering. A simple alternative to recover from reusing the biocatalyst is to provide support with magnetism to separate the biocatalyst from the reaction medium [60,61,62]. PVA supports can also be combined with magnetic materials to improve biocatalyst recovery. This is the case of the immobilization of *B. cepacia* lipase on PVA magnetic supports to catalyze the production of fatty acid ethyl esters [63]. The immobilization methodology consisted first of the preparation of the support, using different magnetization routes that incorporate Fe^+2^ or Fe^+3^ in a SiO_2_/PVA matrix. After synthesizing the SiO_2_/PVA/Magnetic support, this support was activated with epichlorohydrin to immobilize the lipase. The performance of the biocatalyst was tested in the production of fatty acid ethyl esters, reaching a percentage of 96% after 72 h of reaction. This result shows that the biocatalyst was not affected by the effects of the polar compounds present in the reaction medium [63].

Other research reports the immobilization of the lipases of *Burkholderia cepacia* and *Pseudomonas fluorescens* on magnetized matrices of poly(styrene-co-divinylbenzene) formed with PVA (STY-DVB-M) [64]. The biocatalyst preparation consisted of synthesizing the support using PVA, called STY-DVB-M, which was magnetized with 10% magnetite (Fe_2_O_3_) and coprecipitated with Fe^+2^ and Fe^+3^. After synthesizing the support, the physical adsorption method immobilized the lipases. The performance of the biocatalyst was carried out in the catalysis of the ethanolysis of coconut oil at 45 °C for 48 h. The operational stability of both biocatalysts was evaluated by reusing them in seven reaction cycles without decreasing their catalytic activity with a half-life of 970 h [64].

## 6. Concluding and Remarks

The bibliographic analysis of applying the PVA polymer in the immobilization of lipases demonstrates its versatility as a matrix to improve the stability and recoverability of the biocatalysts from the reaction medium. Its high degree of functionalization allows it to be combined with other biodegradable polymers, such as alginate and chitosan, and with different clays to improve the catalytic properties of lipases.

The possibility of forming fibers, membranes, and spheres with high mechanical stability opens a range of options for its application in different types of enzymatic bioreactors, such as membrane bioreactors and stirred tanks, and packed bed bioreactors.

From the point of view of immobilization, this can be carried out by entrapment and by covalent binding, increasing the activity and stability of the enzyme, and an interesting aspect is the compatibility and mechanical stability of PVA supports in the presence of organic solvents and deep eutectic solvents in esterification and transesterification reactions with lipases. PVA-based supports for lipase immobilization using the entrapment methodology have advantages over other polymers because they have high mechanical stability, can be scaled, and can be functionalized and hybridized with other materials.

The ease in scaling immobilization methodologies is important since it makes them attractive for industrial commercialization. PVA supports have many advantages in this direction, alone (like LentiKats^®^) or combined with other materials. They can reach this category and contribute to developing new technologies.

## Figures and Tables

**Figure 1 polymers-15-02021-f001:**
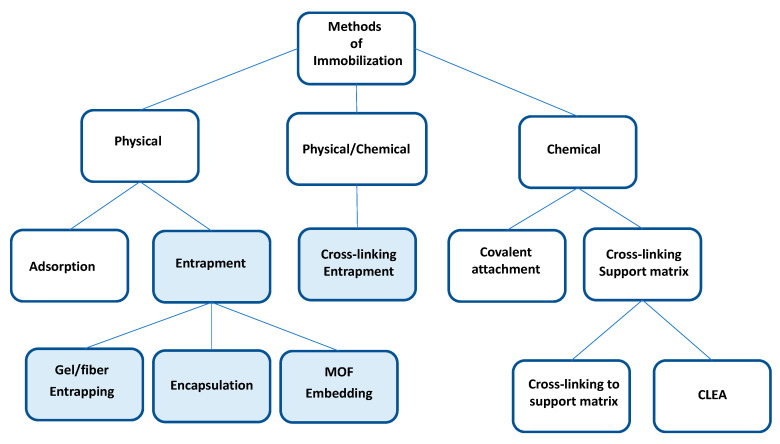
Methods of immobilization. The light blue squares indicate the methodologies where polyvinyl alcohol is used.

**Figure 2 polymers-15-02021-f002:**
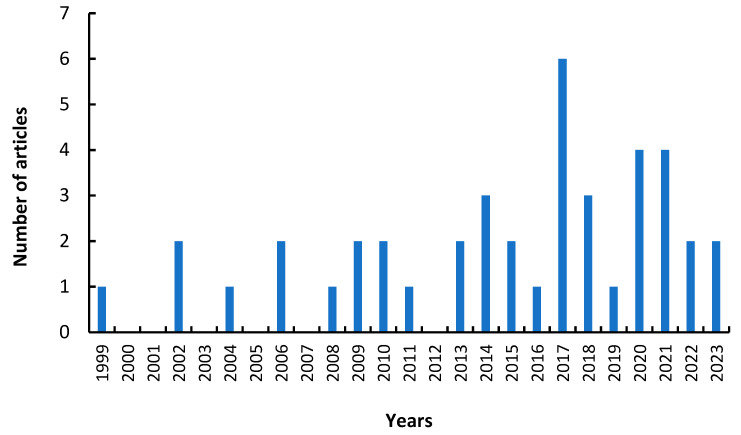
Articles related to the immobilization of lipases on polyvinyl alcohol support by years.

**Figure 3 polymers-15-02021-f003:**
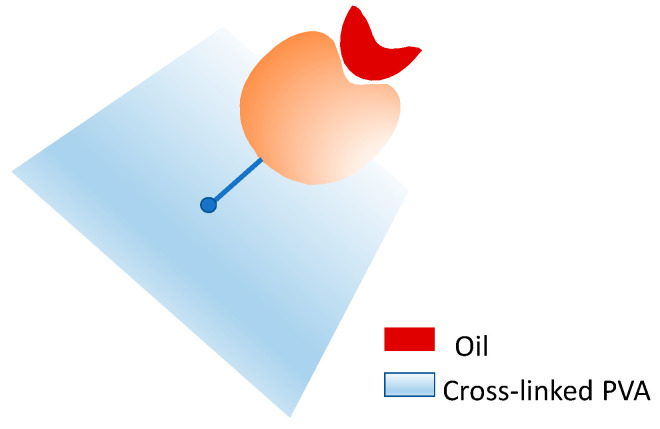
Covalent immobilization of lipase on crosslinked PVA using oils for the stabilization of the open conformation of the active site of the enzyme.

**Figure 4 polymers-15-02021-f004:**
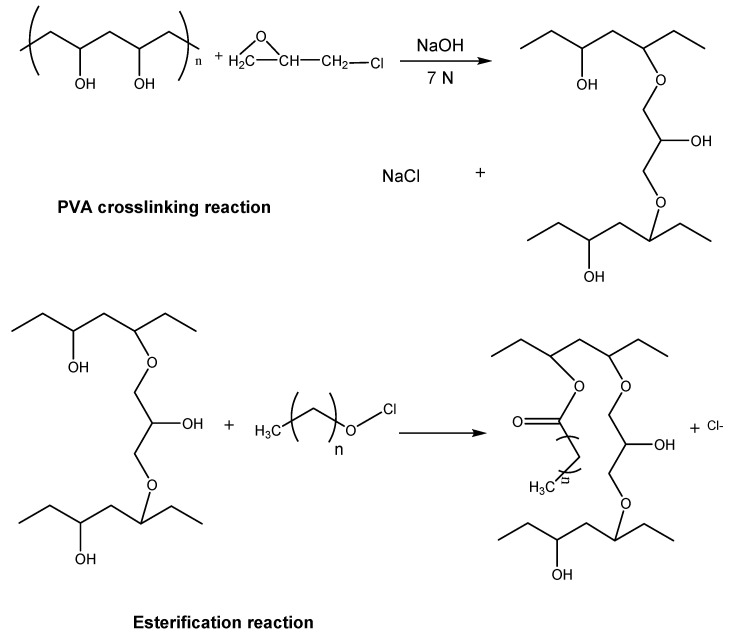
Two-step reaction for the synthesis of esters of (*Cn*-CL-PVA).

**Figure 5 polymers-15-02021-f005:**
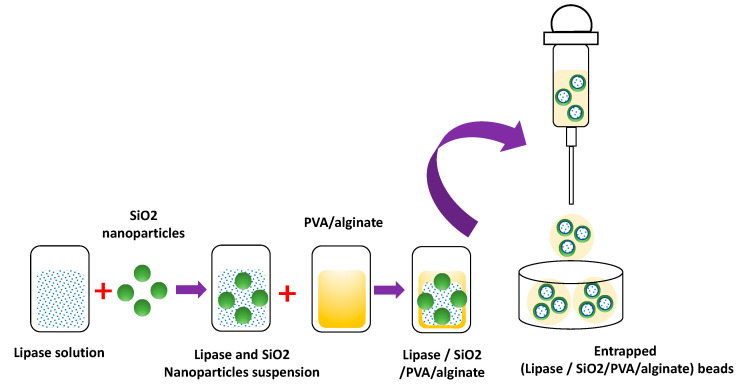
Immobilization of lipase on SiO_2_ nanoparticles trapped in PVA/alginate beads.

**Figure 6 polymers-15-02021-f006:**
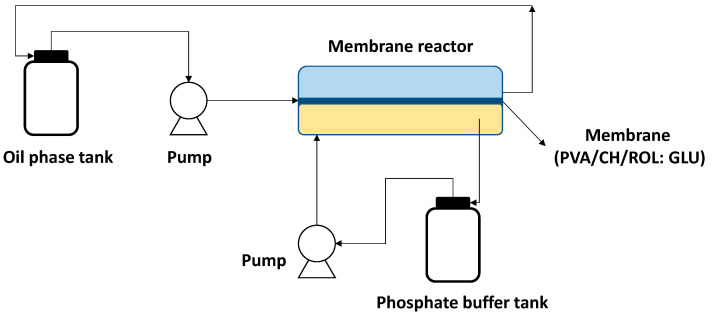
Synthesis of monoglycerides with lipase membrane reactor.

**Figure 7 polymers-15-02021-f007:**
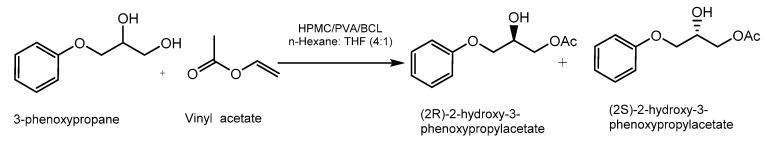
Kinetic resolution of 3- phenoxy-propane-1,2-diols catalyzed by BCL lipase immobilized on PVA/HPMC support.

**Figure 8 polymers-15-02021-f008:**
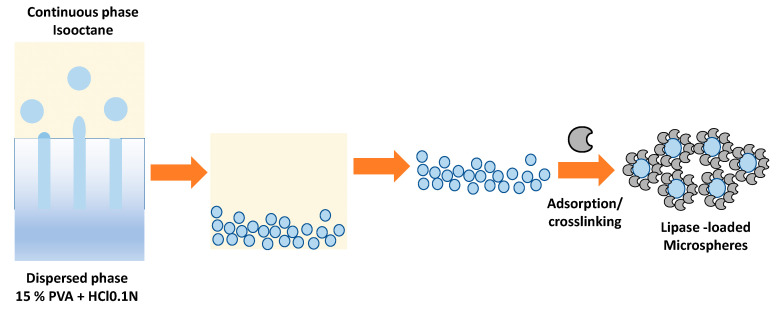
Lipase is immobilized in PVA microspheres through a membrane emulsification process followed by crosslinking with glutaraldehyde (GA).

**Table 1 polymers-15-02021-t001:** Advantages and disadvantages of the immobilization methodologies and support most used for immobilizing lipases.

Support Types or Immobilization Types	Advantages	Disadvantages	Binding
Crosslinking enzyme aggregates (CLEA)	High stability and high activity	Low mechanical resistance	Covalent
Immobead 150 P (Co-polymer of methacrylate)	High mechanical resistance, high stability, scalability	Low activity	Covalent
Accurel MP 1000	High activity, scalability	Leaching	Adsorption
Lewatit VP OC 1600	High activity, scalability	Leaching	Adsorption
Alginate	Avoids agglomeration, biodegradable	Low mechanical resistance	Entrapment
Poly(vinyl) alcohol	High mechanical resistance, scalability	Mass transfer problems	Entrapment
Chitosan	High degree of functionalization, Biodegradable, high stability	Mass transfer problems, low activity	Entrapment, covalent

## Data Availability

Data available in a publicly accessible.

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
