# Peer review of "Immobilization of Lipases Using Poly(vinyl) Alcohol"

_polymers, 2023, doi:10.3390/polym15092021_

Round 1

Reviewer 1 Report

1. What is the main question addressed by the research?

The manuscript shows a review of the possibilities of polyvinyl alcohol use to immobilize lipases when combined with other materials, for example, alginate and chitosan. This issue is essential for advancing the frontier of knowledge in enzyme immobilization.

2. Do you consider the topic original or relevant in the field? Does it address a specific gap in the field?

Yes, I consider it. The manuscript addresses a specific gap in enzyme immobilization through papers that bring methods that use polyvinyl alcohol. Immobilization of enzymes is fundamental to expand the applications of enzymes by enabling the separation and purification of products from reaction mixtures and efficient recovery of enzyme proteins.

3. What does it add to the subject area compared with other published material?

The other published materials present specific experimental procedures with enzymes and different matrixes for immobilization. This manuscript brings several possibilities for PVA use with other materials. For example, “Immobilization of lipase enzyme in polyvinyl alcohol (PVA) nanofibrous membranes (https://doi.org/10.1016/j.memsci.2007.10.008)” and “Immobilization of cellulase in nanofibrous PVA membranes by electrospinning (https://doi.org/10.1016/j.memsci.2004.10.024)”.

4. What specific improvements should the authors consider regarding the methodology? What further controls should be considered?

The author should include in the manuscript the following information:

What were the scientific databases? For example, Web of Science, Scopus, both, or other.

What was the method for selecting manuscripts?

What was the range of years used?

What were the keywords? 

How many papers did the author find? I suggest showing this information in graphic format (y-axis quantity of papers and x-axis years of publication). 

5. Are the conclusions consistent with the evidence and arguments presented and do they address the main question posed?

Yes, the conclusion is consistent with the evidence and arguments.

6. Are the references appropriate?

Yes, references are appropriate and current. The author includes papers between 1999 and 2023. More than twenty papers are from 2020 - 2023.

7. Please include any additional comments on the tables and figures.

The Figures are appropriate, except in Figure 5, where it is necessary to include a legend of the numbers inside the flowchart, not only in the Figure title.

The paper does not include tables. Thus, I suggested including a Table showing the main advantages and disadvantages of the different supports for Immobilization. 

The paper shows a review of the possibilities of the polyvinyl alcohol application as a material to immobilize lipases when combined with alginate, chitosan, and hydroxypropyl methylcellulose. So, it brings an essential contribution to the biotechnology sector. However, before consideration for publication, some points must be reviewed.    

1) Abstract: The text should be rewritten, including the following information: platforms used to analyze existing knowledge; method for selecting manuscripts; quantity of papers found; the range of years used and vacancies in the area. 

2) Abstract: In your opinion, what is necessary to expand the knowledge in this area?

3) Standard the writing of the term Cross-linking because in the text, two forms were found (Cross-linking and crosslinking).

 4) In Figure 5, it would be more didactic to include a legend of the numbers inside the flowchart, not only in the Figure title.

 5) The author could include a Table showing the main advantages and disadvantages of the different supports for Immobilization.

Author Response

We would sincerely like to thank the editor and the reviewer for the careful revision of the paper that has permitted us to improve the overall quality of the manuscript.

Reviewer #1:

  1. What is the main question addressed by the research?

The manuscript shows a review of the possibilities of polyvinyl alcohol use to immobilize lipases when combined with other materials, for example, alginate and chitosan. This issue is essential for advancing the frontier of knowledge in enzyme immobilization.

  1. Do you consider the topic original or relevant in the field? Does it address a specific gap in the field?

Yes, I consider it. The manuscript addresses a specific gap in enzyme immobilization through papers that bring methods that use polyvinyl alcohol. Immobilization of enzymes is fundamental to expand the applications of enzymes by enabling the separation and purification of products from reaction mixtures and efficient recovery of enzyme proteins.

  1. What does it add to the subject area compared with other published material?

The other published materials present specific experimental procedures with enzymes and different matrixes for immobilization. This manuscript brings several possibilities for PVA use with other materials. For example, “Immobilization of lipase enzyme in polyvinyl alcohol (PVA) nanofibrous membranes (https://doi.org/10.1016/j.memsci.2007.10.008)” and “Immobilization of cellulase in nanofibrous PVA membranes by electrospinning (https://doi.org/10.1016/j.memsci.2004.10.024)”.

  1. What specific improvements should the authors consider regarding the methodology? What further controls should be considered?

The author should include in the manuscript the following information:

What were the scientific databases? For example, Web of Science, Scopus, both, or other.

What was the method for selecting manuscripts?

What was the range of years used?

What were the keywords? 

How many papers did the author find? I suggest showing this information in graphic format (y-axis quantity of papers and x-axis years of publication). 

R/ The number of articles per year related to the immobilization of lipases on PVA supports was incorporated in Figure 2 and it was indicated how the information was sought and how many articles related to the subject were found. Please see page 2 lines 77-83.

  1. Are the conclusions consistent with the evidence and arguments presented and do they address the main question posed?

Yes, the conclusion is consistent with the evidence and arguments.

  1. Are the references appropriate?

Yes, references are appropriate and current. The author includes papers between 1999 and 2023. More than twenty papers are from 2020 - 2023.

  1. Please include any additional comments on the tables and figures.

The Figures are appropriate, except in Figure 5, where it is necessary to include a legend of the numbers inside the flowchart, not only in the Figure title.

The paper does not include tables. Thus, I suggested including a Table showing the main advantages and disadvantages of the different supports for Immobilization. 

R/ Figure 5 was modified and Table 1 was added in the introduction pointing out the advantages and disadvantages of the supports most used in the immobilization of lipases.

The paper shows a review of the possibilities of the polyvinyl alcohol application as a material to immobilize lipases when combined with alginate, chitosan, and hydroxypropyl methylcellulose. So, it brings an essential contribution to the biotechnology sector. However, before consideration for publication, some points must be reviewed.  

  • Abstract: The text should be rewritten, including the following information: platforms used to analyze existing knowledge; method for selecting manuscripts; quantity of papers found; the range of years used and vacancies in the area. 

R/ As the reviewer noted, the abstract was rewritten incorporating the information the reviewer suggested on page 1, lines 16-17.

  • Abstract: In your opinion, what is necessary to expand the knowledge in this area?

R/ This Information was incorporated on page 1, lines 17-19.

  • Standard the writing of the term Cross-linking because in the text, two forms were found (Cross-linking and crosslinking).

R/ The word "crosslinking" was standardized throughout the manuscript.

  • In Figure 5, it would be more didactic to include a legend of the numbers inside the flowchart, not only in the Figure title.

R/ Figure 5 was modified.

  • The author could include a Table showing the main advantages and disadvantages of the different supports for Immobilization.

R/ As noted by the reviewer, Table 1 was included in the introduction section.

Reviewer 2 Report

This is an interesting review about the immobilization of lipases using poly(vinyl) alcohol. I suggest it for publication after the following points are addressed.

1. The font size in the figure should be bigger.

2. Line 43-44, these sentences should be incorporated into the last paragraph.

3. Line 35-36, one study (Journal of Materials Chemistry B 1 (19), 2482-2488) should be included to support such a claim.

4. The perspective of this review should be expanded.

5. The authors should discuss why PVA is more suitable for enzyme immobilization less other polymers.

Minor editing of English language required

Author Response

We would sincerely like to thank the editor and the reviewer for the careful revision of the paper that has permitted us to improve the overall quality of the manuscript.

Reviewer #2:

  1. The font size in the figure should be bigger.

R/ The font size of whole figures was modified.

  1. Line 43-44, these sentences should be incorporated into the last paragraph.

R/ As noted by the reviewer, the sentence was incorporated into the last paragraph

  1. Line 35-36, one study (Journal of Materials Chemistry B 1 (19), 2482-2488) should be included to support such a claim.

R/ The study in Journal of Materials Chemistry B 1 (19), 2482-2488 was incorporated.

  1. The perspective of this review should be expanded.

R/ This review covers complete information on the applications of PVA supports in the immobilization of lipases from the beginning of its application around the year 1999 to date. The information search methodology and the number of articles per year were included to broaden the perspective. Please see page 3, lines 86-88, and Figure 2.

  1. The authors should discuss why PVA is more suitable for enzyme immobilization less other polymers.

R/ PVA-based supports for lipase immobilization using the entrapment methodology have advantages over other polymers because they have high mechanical stability, can be scaled, and can be functionalized and hybridized with other polymers. Please see page 12, lines 439-441.

Reviewer 3 Report

The presented review is devoted to the problem of using lipases. The authors briefly but clearly describe the possible ways of using lipases, their advantages and disadvantages. Next, the authors concentrate on the use of PVA to immobilize lipases. The possibilities of introducing into the system biopolymers such as chitosan, alginate, etc. are considered. Examples of the practical application of systems in the form of molded materials - fibers, membranes, etc. are given. In my opinion, the presented work was done at a decent level. However, before publishing the review, it is necessary to make minor changes to it. Line 15. "well as the incorporation of magnetic properties" - not clear statement The list of keywords needs to be expanded. Lines 154, 158. Replace "four" with "4". Line 181. "(, " - needs to be corrected. Figure 5. In the figure caption, the decoding for - 4 is missing.   As a recommendation, I would like to suggest adding photographic materials to the work (photographs of the materials obtained, their morphology (SEM), etc.). I would also like to see information about the structural features of the materials obtained.

Ok

Author Response

We would sincerely like to thank the editor and the reviewer for the careful revision of the paper that has permitted us to improve the overall quality of the manuscript.

Reviewer #3:

The presented review is devoted to the problem of using lipases. The authors briefly but clearly describe the possible ways of using lipases, their advantages and disadvantages. Next, the authors concentrate on the use of PVA to immobilize lipases. The possibilities of introducing into the system biopolymers such as chitosan, alginate, etc. are considered. Examples of the practical application of systems in the form of molded materials - fibers, membranes, etc. are given. In my opinion, the presented work was done at a decent level. However, before publishing the review, it is necessary to make minor changes to it.

  • Line 15. "well as the incorporation of magnetic properties" - not clear statement

R/ The sentence was corrected. Please, see line 15.

  • The list of keywords needs to be expanded.

R/ The list of keywords was expanded. Please, see lines 20-22

  • Lines 154, 158. Replace "four" with "4".

R/ The word was replaced by the number 4. Please, see line 175.

  • Line 181. "(, " - needs to be corrected.

R/ The “(“ was deleted.

  • Figure 5. In the figure caption, the decoding for - 4 is missing.

R/ The mistake was corrected.

  • As a recommendation, I would like to suggest adding photographic materials to the work (photographs of the materials obtained, their morphology (SEM), etc.). I would also like to see information about the structural features of the materials obtained.

R/ Due to the rights and permissions, they make it difficult to reproduce the original images that the articles have, which readers can access through the reference in the original article. However, an explanation of the differences between the support alone and with the immobilized enzyme by XRD characterization and SEM was included in lines 322 and 325 of the manuscript.

“The characterization of the support and the immobilized lipase, this was carried out by means of X-ray diffraction (XRD) and Scanning Electron Microscope (SEM). Through XRD it was possible to distinguish a peak belonging to the presence of lipase and through SEM, it was possible to observe the presence of the enzyme in the matrix.”

Round 2

Reviewer 1 Report

The manuscript has been sufficiently improved to warrant publication in Polymers.